# The Impact of Salt Concentration on the Mineral Nutrition of *Tetragonia tetragonioides*

**Gulom Bekmirzaev [1],*, Baghdad Ouddane [2] , Jose Beltrao [3] and Yoshiharu Fujii [4]**

[1]  Department of Irrigation and Melioration, Tashkent Institute of Irrigation and Agricultural Mechanization Engineers, Tashkent 100000, Uzbekistan

[2]  Physico-Chemistry Team of the Environment, Sciences and Technologies, University of Lille, LASIR UMR-CNRS 8516, Building C8, 59655 Villeneuve d'Ascq, CEDEX, France; baghdad.ouddane@univ-lille1.fr

[3]  Research Centre for Spatial and Organizational Dynamics, University of Algarve, Campus de Gambelas, 8005-139 Faro, Portugal; jbeltrao@ualg.pt

[4]  Tokyo University of Agriculture and Technology, International Environmental and Agricultural Sciences, Fuchu Campus 3-5-8, Saiwai-cho, Fuchu, Tokyo 183-8509, Japan; yfujii@cc.tuat.ac.jp

*  Correspondence: gulombek@gmail.com

**Abstract:** The purpose of the experiment was to study the effect of salinity (NaCl) on growth, biomass production (total yield), mineral composition (macro- and micronutrient contents in leaves and the soil in which the plant is grown) of *Tetragonia tetragonioides* during the vegetation period. The experimental work was conducted in the greenhouse at the University of Lille 1, France, from 2 November 2015 to 25 January 2016. Three salinity treatments (T1 (50 mM NaCl), T2 (100 mM NaCl), T3 (200 mM NaCl)) and a control treatment (T0 (0 mM NaCl)) were applied. Analysis of the results showed that the total yield of the crop had low variation between the salinity treatments and the control treatment. The salt concentrations had an effect on the macro- and micronutrient contents in leaves and soil. In conclusion, *T. tetragonioides* exhibited good potential for use as a species to remove salt. This is the main important finding of this research.

**Keywords:** salinity; biomass production; macro- and micronutrients; salt removal; total yield

## 1. Introduction

Salinization of soils is a well-known problem often associated with anthropogenic factors as well as climate changes. Salinity seriously threatens the provision of high-quality agricultural products. The increase in the population during recent years implies an increase in agriculture and, consequently, a greater consumption of water, pushing this resource to its limits [1]. Agricultural practices (i.e., fertilizer application, poor irrigation systems coupled with increasing sea levels) have led to poor water quality and saline soil conditions [2]. Salinity stress affects seed germination, seedling growth, leaf size, shoot growth, shoot and root lengths, shoot dry weight, shoot fresh weight, the number of tillers per plant, flowering stage, spikelet number, percent of sterile florets and productivity [3–7].

Growing salt-tolerant crops can give good results on saline soils [8,9]. To improve the salinity tolerance of crops, various traits can be incorporated, including ion exclusion, osmotic tolerance and tissue tolerance [10]. Saline tolerance in plants may be explained by the involvement of functional and structural adaptations, such as growth regulation, osmotic adjustment, changes in water potential [11], mineral nutrient changes, and hormone balance, all of which can alleviate the harmful effects of these stresses [12]. Plant tolerance to saline irrigation water is generally assessed by growth reduction. In ornamental plants, however, one should also consider the appearance of foliage, because salt stress

can result in leaf damage that reduces their market value [13–15]. Understanding $Na^+$ sensing and transport in plants under salt stress will be of benefit for the robust breeding of salt-tolerant crop species [16]. The potential applications of halophytes in phytoremediation, desalinization, secondary metabolite production, food and saline agriculture have been discussed, and the harvested halophytes can have industrial value; subsequently, rehabilitated soil can be utilized for agricultural purposes [17].

The use of saline water is becoming an important approach to reclaim and utilize salt-affected soil for landscaping and agricultural purposes [18]. Saline waters at six different salinities were applied to spinach (*Spinacia oleracea* L. Matador) grown in pots. Soil salinity increased linearly with increasing salinity of irrigation water [19]. The use of saline water is an option for the irrigation of salt-tolerant ornamentals as competition for high-quality water increases. However, despite the importance of ornamental shrubs in Mediterranean areas, the salt tolerance of such species has received little attention [20]. One of the processes that promote soil salinization is the use of reclaimed irrigation water [21,22]. Saline water has been increasingly used for crop production in areas characterized by freshwater shortages [23]. However, continuous irrigation with saline water often results in adverse effects on the soil–crop system. Irrigation water is one of the major factors determining crop yields and agricultural production [24]. The most fundamental agricultural use for saline water is to provide water for plant absorption [25].

New Zealand spinach (*Tetragonia tetragonioides*) is a perennial plant used in salads; it is suitable for warmer climates and is salt tolerant. New Zealand spinach is reported to have medicinal uses [26]. Greenhouse cultivation is noted for high uptake of minerals, consistent climatic conditions, exclusion of natural precipitation and control of salt accumulation [27]. In the greenhouse industry, methods have been developed for the determination of the nutrient availability and salinity status of the soil and substrate [28].

Understanding the mechanisms of the tolerance of crop plants to high concentrations of NaCl in soils may ultimately help to improve yield on saline lands. Despite the fact that most plants accumulate both sodium ($Na^+$) and chloride ($Cl^-$) ions at high concentrations in their shoot tissues when grown in saline soils, most research on salt tolerance in annual plants has focused on the toxic effects of $Na^+$ accumulation [29]. NaCl is the most common salt that causes salinity stress. Salinity hazards caused by irrigation depend on the type of salt, soil, and climatic conditions, crop species, and the amount, quality, and frequency of water applications [30].

However, saline soils lead to a decrease in crop yields. Therefore, the main goal of this study is to study the negative effects of salinity on plant growth, biomass production (total yield) and the content of minerals (macro- and micronutrients in plant leaves and saline soils).

## 2. Materials and Methods

### 2.1. Experimental Procedure

The experimental study was conducted in the greenhouse at the University of Lille 1, France. Seeds of *T. tetragonioides* were sown in the soil on 2 November 2015. The seeds of the plant germinated after a week (Day 9). Six leafy plants were transplanted into three randomized pots (each contained 1300 g soil$^{-1}$) on day 26.

The plants were irrigated with tap water until the beginning of the salinity treatments. A nitrogen fertigation treatment was the concentrations of 2 mM $NO_3^-$ and 2 mM $NH_4^+$. The treatments (every treatment has four replications) were as follows: salinity treatments were T1 (5–8) 50 mM NaCl, T2 (9–12) 100 mM NaCl and T3 (13–16) 200 mM NaCl and control was T0 (1–4) 0 mM NaCl. The number of plants per treatment was four. The plants were irrigated with saline water at a minimal amount, enough for plant survival (0.25 L pot$^{-1}$ at the beginning of the experiment). The variants with saline water received 0.50 L pot$^{-1}$ until December 17. Measurement of plant germination started two days after transplantation to the randomised pots. The stem length and the number of nodes of the plants were analysed every 10 days during the vegetation period.

After 60 days (25 January 2016) of salinity treatments, four plants from each treatment were collected and washed with distilled water for a few minutes, wiped with paper, and the fresh weight (FW) was measured. The fresh samples were dried in a forced draught oven at 65 °C for 72 h, and the dry weight (DW) was measured, after which the plant materials were collected for chemical analyses.

Analyses of the electrical conductivity ($EC_w$) and pH of the drainage water were carried out every 10 days after the plants were watered (from day 26) for the duration of the experiment. The soil electrical conductivity ($EC_s$) and pH were analysed before and after the experimental study. The root length of the plants was determined at the end of the study.

### 2.2. Soil

The soil for the experiment was used in NEUHAUS, which is generally used for horticultural crops in greenhouses. The soil analyses were conducted at the special laboratory of the LASIR at the University of Lille 1 before the beginning of the experimental study (Table 1). The soil pH was 6, and $EC_s$ was 0.3 (dS m$^{-1}$). Table 1 shows that the mineral composition (macro- and micronutrients) of the soil was generally poor (% g soil$^{-1}$). The soil macronutrient (N, P, K) contents were low and the micronutrient contents were very low.

**Table 1.** Soil parameters and mineral elements (macro- and micronutrients).

| Soil Parameters | | Macronutrients (%) | | | | | |
|---|---|---|---|---|---|---|---|
| pH, $EC_s$ (dS m$^{-1}$) | | N | P | K | Ca | Mg | S |
| 6, 0.3 | | 0.01 | 0.03 | 0.17 | 0.88 | 0.21 | 0.14 |
| Micronutrients (%) | | | | | | | |
| Fe | Al | Sr | Zn | Cu | Pb | Na | Cl |
| 0.34 | 0.18 | 0.002 | 0.002 | 0.001 | 0.001 | 0.03 | 0.01 |

### 2.3. Chemical Analyses

The dried soil and leaf samples were used to analyse the ion concentrations. The dry materials were ground and digested via the dry digestion method [31]. The concentrations of copper ($Cu^{2+}$), iron ($Fe^{2+}$), zinc ($Zn^{2+}$), calcium ($Ca^{2+}$), magnesium ($Mg^{2+}$), potassium ($K^+$), phosphorus (P) and sodium ($Na^+$) were determined by inductively coupled plasma–atomic emission spectrometry (ICP-AES) [32]. After the determination of the ion concentrations, the $K^+/Na^+$ and $Ca^{2+}/Na^+$ ratios were calculated.

Chloride ions ($Cl^-$) were determined in the aqueous extract by titration with silver nitrate according to [33]. Plant nitrogen (N) was determined by the Kjeldahl method [34].

### 2.4. Climate Conditions in the Greenhouse

The average climatic data during the experimental period in the greenhouse were as follows: the maximal relative humidity was 87.9%, and the minimal relative humidity was 13.2%; the maximum temperature was 22.2 °C (November 10) at the start of the experiment; the minimum temperature was 19.9 °C on 30 November and 20 January.

### 2.5. Statistical Analysis

The statistical analysis, including analysis of variance (ANOVA) and Duncan's multiple range tests, was performed to study the significance of different salinity gradients on the different parameters studied. Values were calculated at the $p \leq 0.05$ probability levels. All statistical analyses were conducted using SPSS 21.0 (SPSS Inc., Chicago, IL, USA).

## 3. Results and Discussion

### 3.1. Fresh and Dry Mass of the Plant

Plant growth (stem length and number of nodes and leaves of the plants) was analysed every 10 days during the vegetation period (30 days). The plants developed periodically until the start of the salinity treatments on 17 December. In this research, the plants were exposed to salt stress by increasing the NaCl concentration (0, 50, 100 and 200 mM) of irrigation water. The salinity treatments had a significant effect on the growth rate of the plants. Thus, with an increase in salinity treatments, the plant growth decreased, and this result corresponds with the data [35]. In the present study, the growth of New Zealand spinach increased under salt stress, agreeing with previous data reported for the halophytes *Salicornia europaea*, *Suaeda maritima* [36] and *Alhagi pseudalhagi* [37], in which salt treatment at low levels improved plant growth. These results indicated that New Zealand spinach is a halophyte, so this species is salt tolerant.

The results of the experiment with *T. tetragonioides* showed that the fresh weights in variants with salt decreased inconsistently compared to the control group (from 261 to 326 g plant$^{-1}$ and 315 and 283 g plant$^{-1}$) as they were affected by increased salinity (Figure 1A). The fresh weight in all variants was different. The biomass production of stems (DW g plant$^{-1}$) and the dry weight of leaves (DW g plant$^{-1}$) did not differ between salinity treatments (Figure 1B) and were higher than those of the control variant. The dry weight of seeds (DW g plant$^{-1}$) was lower in variants with salt than in the control variant (T0).

The partitioning among plant organs was affected by the medium salt concentration; namely, under salt conditions, there was an increase in the percentage of dry matter of the stems and leaves and a decrease in the number of seeds, similar to other studies [19,38,39]. The fresh and dry weights of New Zealand spinach increased significantly at a salt concentration of 50 mM and remained unchanged in variants with salt concentrations of 100 and 200 mM compared to the control group. This is similar to the results obtained by [40], who found an increased yield of salt-tolerant species (New Zealand spinach).

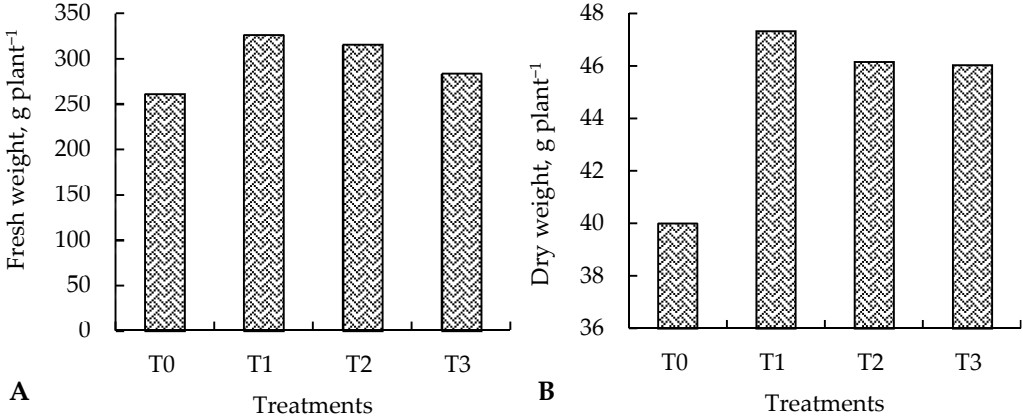

**Figure 1.** Fresh (**A**) and dry (**B**) weights of the plants.

### 3.2. Electrical Conductivity of Drainage Water, pH and EC$_s$ of the Soil after the Experiment

Table 2 shows that the electrical conductivity (EC$_w$) of drainage water was analysed for each treatment during the vegetation period. The EC$_w$ of drainage water slightly increased until the next irrigation event of the plants in the salinity treatments, whereas that of T0 (control) decreased. The values were constant in all treatments in the last two irrigation events of the vegetation period.

The pH and electrical conductivity (EC$_s$) of a dissolved soil sample were analysed at the end of the experiment by the EC meter (Table 2). Table 2 shows that the pH of the soil decreased with an increase in the salt concentration. The soil was alkaline. The EC$_s$ of the soil increased with an increase in the salt concentration.

**Table 2.** Electrical conductivity ($EC_w$) of the drainage water, pH and $EC_s$ of the soil at the end of the experiment. Different letters within a column represent significant differences ($p \leq 0.05$).

| Treatment | $EC_w$ of the Drainage Water, $EC_w$, mS/cm | | Soil Parameters (End of Exp.) | |
|---|---|---|---|---|
| | 27.11.2015 | 17.01.2016 | pH | $EC_s$ (dS m$^{-1}$) |
| T0 | 2.4 ± 0.23 c | 2.4 ± 0.06 d | 5.5 ± 0.29 a | 1.0 ± 0.00 c |
| T1 | 2.5 ± 0.31 c | 6.8 ± 1.02 c | 5.0 ± 0.0 ab | 2.75 ± 0.48 bc |
| T2 | 2.7 ± 0.44 c | 12.7 ± 2.73 b | 4.8 ± 0.25 b | 3.75 ± 0.63 b |
| T3 | 2.7 ± 0.42 c | 20.2 ± 4.24 a | 5.0 ± 0.0 ab | 7.25 ± 1.11 a |

### 3.3. Macro- and Micronutrients in the Leaves of Plants

The relationship between the degree of salinization and the mineral nutrition of plants is extremely complex. Salinization that adversely affects plant growth causes nutritional disorders that may result from the effects of salinity on nutrient availability, absorption and transport in plant organs [41]. Trace elements such as nitrogen (N), phosphorus (P) and sulphur (S) in the leaves were lower in each variant (Figure 2A). In an experimental study, salt stress reduced $K^+$ absorption and increased the $Na^+/K^+$ ratio. Concentrations of $Ca^{2+}$ and $Mg^{2+}$ in New Zealand spinach leaves decreased in variants with salt. An increase in $Na^+$ in plants was associated with reduced $K^+$, $Mg^{2+}$, and $Ca^{2+}$, indicating that the absorption of these macronutrients was limited, as noted for other halophytic plants [42,43].

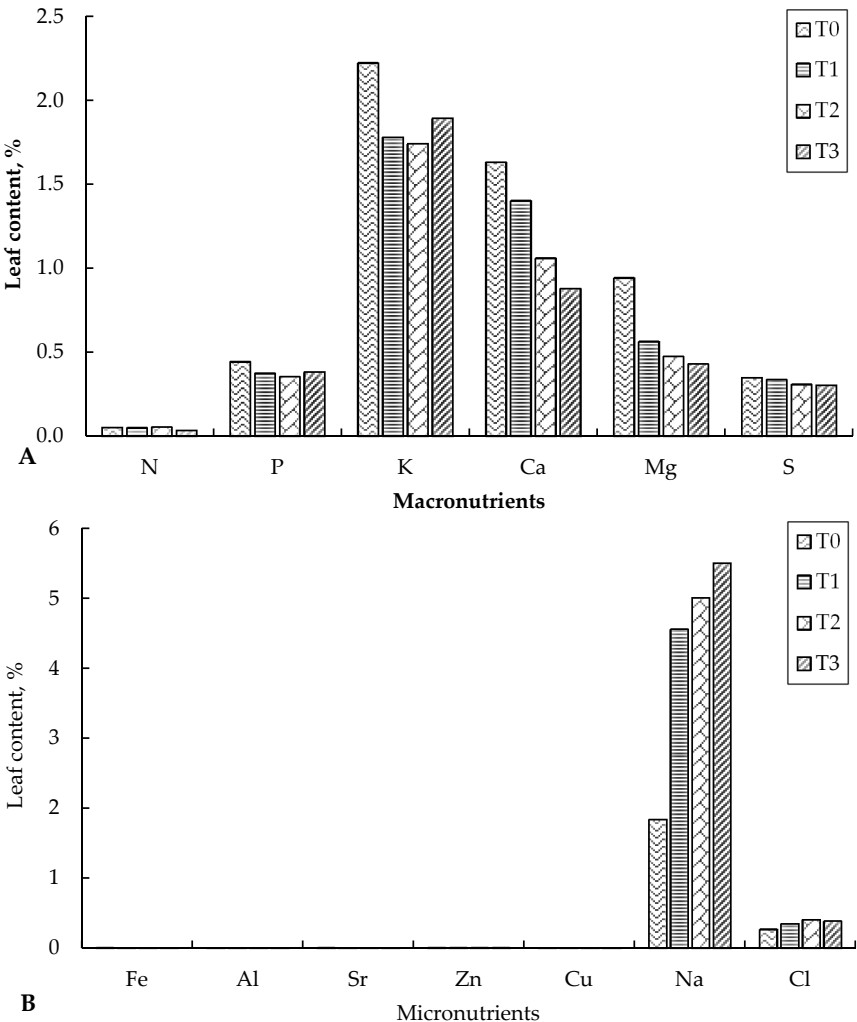

**Figure 2.** Contents of macro- (**A**) and microelements (**B**) in leaves.

Figure 2B shows that the trace elements (Fe, Al, Sr, Zn and Cu) in the leaves of the plants had a very low concentration. As noted in many reports, halophytic plants accumulate and store large amounts of $Na^+$ in the vacuoles to reduce the osmotic potential [44–47]. Surprisingly, chlorine ($Cl^-$) had a low concentration in the leaves, and there were no differences between salt treatments.

### 3.4. Contents of Macro- and Micronutrients in Soil

Among plant macronutrients, potassium plays an important role in mitigating the adverse effects of a high salt content in soil. It also helps to retain plant water. Adequate levels of potassium in plants also allow the roots to absorb/extract water from the soil, even under low humidity conditions. Soil macronutrients (N, P, K) are essential for plant growth and, based on comparisons with [48], had a very low concentration (0.03–0.07%) in all variants (Figure 3A). The use of potash fertilizers increases the movement of K from the soil to the root system. Phosphorus, $K^+$ and nitrogen (N) are important macronutrients that are involved in many important functions in the plant, especially during storage and energy transfer. The interaction of phosphorus (P) with salt in plant nutrition is very complex and sometimes confused depending on the plant type, growth stage, salt, degree of salinity and P content during development [49]. Calcium ($Ca^{2+}$) had a larger share in the soil compared to other macronutrients. The magnesium ($Mg^{2+}$) and sulphur (S) in the soil contents ranged from 0.41–0.47% among the variants.

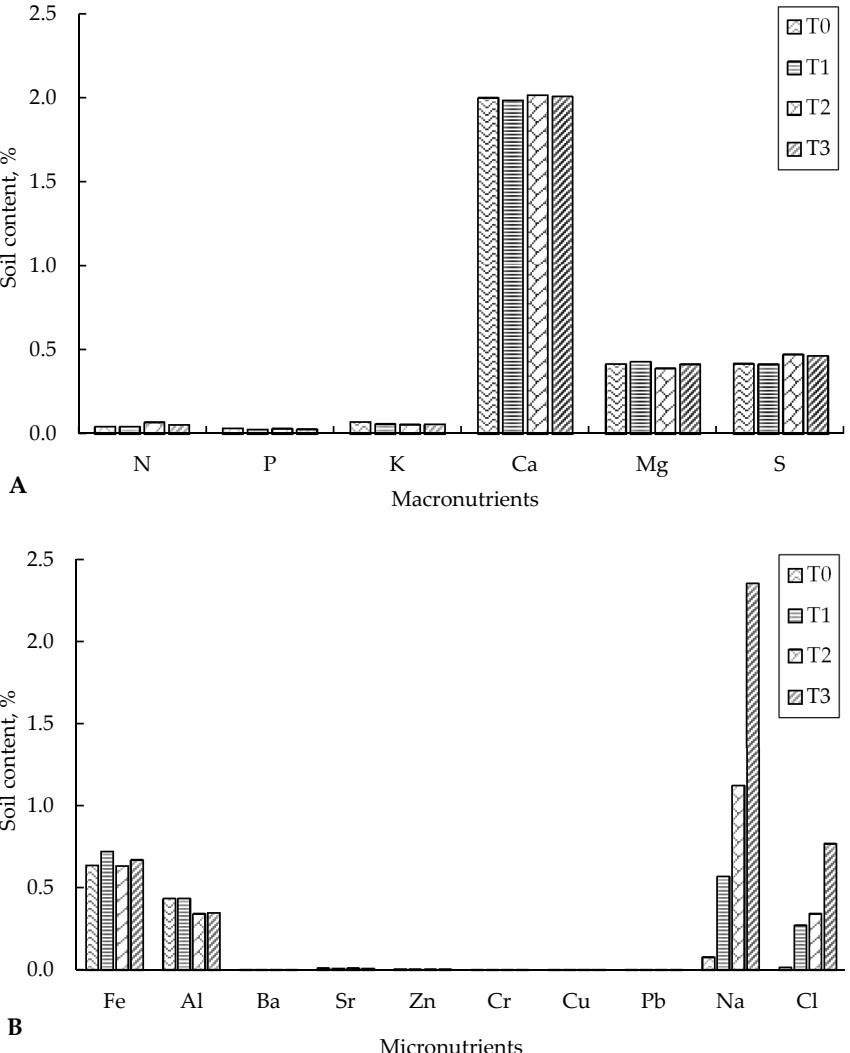

**Figure 3.** Macro- (**A**) and micronutrients (**B**) in soil.

The content of micronutrients in the soil with New Zealand spinach (Figure 3B) differed among the variants with salt. In the soil of each variant, there were low concentrations of iron (Fe) and aluminium (Al), i.e., 0.34–0.72%. The microelements Ba, Sr, Zn, Cr, Cu and Pb in the soil had very low concentrations (0.001–0.01%) [50]. The high degree of salinization had a clear effect on trace elements such as sodium (Na) and chlorine (Cl) in the soil.

*3.5. Yield of Plants*

Fresh leaves and stems are consumable parts; therefore, weight is an important component of yield, as well as plant height, which are positively correlated factors in fresh crops [8]. The average crop weight was 326 (FW) (T1), 316 (FW) (T2), 284 (FW) (T3), and 261 g plant$^{-1}$ (FW) (T0) (Table 3).

**Table 3.** Yield of plants. Different letters within a column represent significant differences ($p \leq 0.05$).

| Treatment | FW (g plant$^{-1}$) | DW (g plant$^{-1}$) | Yield % | FM kg ha$^{-1}$ | DM kg ha$^{-1}$ |
|---|---|---|---|---|---|
| T0 | 261 ± 6.2 b | 40 ± 1.5 b | 15.3 ± 0.3ab | 31329 ± 748 b | 4800 ± 177 b |
| T1 | 326 ± 12.1 a | 47 ± 2.4 a | 14.8 ± 0.3ab | 39114 ± 1454a | 5679 ± 287 a |
| T2 | 316 ± 8.3 a | 46 ± 2.7 ab | 14.5 ± 0.7 b | 37851 ± 1008a | 5538 ± 316 ab |
| T3 | 284 ± 6.4 b | 46 ± 1.4 ab | 16.3 ± 0.5 a | 34008 ± 751 b | 5523 ± 149 ab |

New Zealand spinach produced a significant amount of dry matter, which ranged from 40 to 47 g plant$^{-1}$ and from 4800 to 5679 kg ha$^{-1}$ (DM). The allocation of plant dry matter among plant organs changed as an effect of the high salt concentration (NaCl). Unexpectedly, an increase in the dry matter content was observed in variants with salt (T1, T2, T3), and there was a decrease in dry matter in the control group (T0).

*3.6. Ion Extraction of Plants*

*T. tetragonioides* showed high potential salt (ion) removal (Table 4). *Tetragonia tetragonioides* can remove salts from approximately 20–25 cm of the topsoil layer. The plants removed up to 500 kg ha$^{-1}$ of ions. Previous investigations in the Mediterranean area have shown that annual *T. tetragonioides* crops produced the highest biomass and were efficient crops to remove ions from salt-affected soils. For instance, [27] found that *T. tetragonioides* produced 4200 DM kg ha$^{-1}$ and removed up to 700 kg ha$^{-1}$ NaCl in Portugal. Therefore, in future investigations, we need to assess plants such as *T. tetragonioides*, for their efficiencies removing ions from saline soils.

**Table 4.** Ion extraction from the soil by *Tetragonia tetragonioides*, (Na) kg ha$^{-1}$.

| Treatment | Area m$^2$ | Plant Density | FW g plant$^{-1}$ | DW g plant$^{-1}$ | FY kg ha$^{-1}$ | DY kg ha$^{-1}$ | mg g$^{-1}$ | Ion ext. mg plant$^{-1}$ | g m$^{-2}$ | Ion ext. kg ha$^{-1}$ |
|---|---|---|---|---|---|---|---|---|---|---|
| T0 | 1 | 12 | 261 | 40 | 31,329 | 4800 | 18.3 | 729.3 | 8.8 | 42.0 |
| T1 | 1 | 12 | 326 | 47 | 39,114 | 5679 | 45.6 | 2156.5 | 25.9 | 148.1 |
| T2 | 1 | 12 | 316 | 46 | 37,851 | 5538 | 50.1 | 2302.9 | 27.6 | 154.2 |
| T3 | 1 | 12 | 284 | 46 | 34,008 | 5523 | 54.9 | 2527.5 | 30.3 | 167.7 |

## 4. Conclusions

The study results showed effects of the salinity on the growth, biomass productions (yield), mineral compositions (macro- and micronutrient contents in leaves and the soil which plant is grown) of *Tetragonia tetragonioides*.

Plant growth (stem length, number of nodes and leaves of the plant) was analysed and the plants developed periodically until the start of the salinity treatments. The salinity had a significant effect

on the growth rate of the plants. Thus, with an increase in salinity treatments, the plant growth decreased. The results of the experiment with *T. tetragonioides* showed that the fresh weights in the salinity treatments with salt decreased inconsistently compared to the control as they were affected by increased salinity. The fresh weight in all treatments was different.

The biomass production of the plants (stems and leaves) did not differ between salinity treatments and were higher than those of the control. The dry weight of seeds was lower in the salinity treatments than in the control. There was an increase in the percentage of dry matter of the stems and leaves and a decrease in the number of seeds under salt conditions. The fresh and dry weights of the plants increased significantly at a salt concentration of 50 mM NaCl. In conclusion, the growth of *T. tetragonioides* was promoted under saline conditions, indicating that this species is halophytic.

Analysis of the results showed that the total yield of the plant had low variation between the salinity treatments and the control. The salt concentrations had an effect on the macro- and micronutrient contents in leaves and soil. In conclusion, it can be suggested that New Zealand spinach is resistant to high salinity conditions. The plant turned out to be a species with good potential for the removal of salts (ions). As a final remark, it is concluded that under salinity conditions, clean and environmentally safe procedures to control salinity could be associated with conventional techniques, combining environmental and social aspects. Hence, the species may contribute to increasing the soil sustainability of irrigated areas and may also be used as a food crop.

**Author Contributions:** The paper is the result of the collaboration among all authors; however, G.B. and J.B. contributed to the all sections. B.O. contributed to the sections on macro- and micronutrients in leaves and soil contents, ion extraction of plants. Y.F. contributed on the section of electrical conductivity of drainage water, pH and EC$_s$ of the soil. All authors have read and agreed to the published version of the manuscript.

**Funding:** This research was partially supported by JST CREST Grant Number JPMJCR17O2.

**Acknowledgments:** We are grateful to the Erasmus Mundus Euro-Asian CEA project for support of the European Joint Doctorate Program and for technical maintenance for several analyses conducted in the LASIR laboratory at the University of Lille 1, France.

**Conflicts of Interest:** The authors declare no conflict of interest.

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
