# Peer review of "The Impact of Salt Concentration on the Mineral Nutrition of Tetragonia tetragonioides"

_agriculture, doi:10.3390/agriculture10060238_

Round 1

Reviewer 1 Report

This manuscript deals with salt stress to New Zealand spinach and the potential side effects of increasing salt to spinach growth and chemical composition. This manuscript has the potential to be a nice summary of salt stress on the plant, however as is, it is largely a list of facts and data.  To improve both the narrative and the conclusions of the study, I strongly suggest the results/discussion section to be split into two distinct sections. This will enable the facts to remain as they are presented, while drawing attention to the discussion. Discussion of the results is largely missing, and as such, I cannot provide a meaningful review of the bulk of the manuscript.  Currently, conclusions are drawn, which although stem from the results, there is no connection between raw data and conclusions. I think the authors will see that once the results are separated into their own section, little text is left that actually discusses the results, compares and contrasts with the literature, or draws conclusions about the text. 

To help tell the story in the discussion section, it may be necessary to better outline the aims of the manuscript and study. This can be done in the last paragraph of the introduction, where a statement stating the aims is missing.  The reader can infer the aims of the study through that last paragraph, but may not be accurate. It reads as if the addition of Cl- will be tested, as the effects of Na+ are well known, although the study does not tease apart the effects of Na+ vs Cl-. 

Specific comments are below-

Line 28-36. Integrate citations better within the paragraph.  Several statements are made without appropriate citations to back them up. In fact, throughout the introduction, supporting citations are missing or are underrepresented. This can be improved by increasing the number of citations throughout the introduction or citing more than one study per sentence. This is especially true when making a broad statement, such as L58-59, which has no citation but says “often results,” with no citations in support.

Methods: The methods section is lacking a coherent order and flow.  It reads as if sections are inserted randomly with no regard to the order of the experiment. The sections break up the methods nicely, but the order of each section could be improved. This is especially true for sections 2.4 and 2.5.

Dates could be better represented by Day numbers (Day 7, Day 30) rather than dates. Specific dates would be important if this were an outdoor study, but as it was inside a greenhouse, Day numbers are sufficient.

Line 101. Dry digestion method needs a citation.

Line 106. The Kjeldahl method needs a citation.

Section 2.5. Reconsider whether the soil and table discussion should belong in the results rather than the methods.

Figures 2 and 3. The line graphs should instead be bar graphs. The x axis is categorical, and there is no logical connection between categories. A line graph is used to infer data between the distinct points, therefore, a bar graph should be used.  Furthermore, the spline function fitted to the graph presents negative leaf and soil content in between Zn and Cu (Figure 2a) and P and K (Figure 3a), respectively.  For the bar graphs that are used, there is no error term, and the reader cannot tell the spread of the data and how significant the differences between treatments are.

Line 131. The citations of [30] is out of place.

Line 226-227. You cannot say this spinach species has the greatest potential for salt removal, as you only tested one species. Same with different variants, only one variant was used. 

Author Response

Dear Reviewer,

Thank you so much for your time and your suggestions for the paper.

Best regards,

Gulom Bekmirzaev

Reviewer 2 Report

Paper entitled “The Impact of Salt Concentration on the Mineral Nutrition of Tetragonia tetragonioides” is an interesting paper, considering that the salinity is affecting more and more once productive soils worldwide.

The findings of this study may prove to be useful for the saline agriculture management because the plant species studied showed a great potential for the removal of salts, while it may be used for feed or even food (so, maybe inter-cropping with Tetragonia tetragonioides is not a bad idea when growing crops in a saline soil).

Small suggestions may be found below as follows:

L 17: “… leaves and soil) of Tetragonia tetragonioides” – soil is not really fitting here as it is not part of the plant per se, so, although I’m aware of word limitation, I still suggest some reformulation of the sentence…

L 77: I think some extra space is appearing at the end of line

L 84: Maybe it would be better to start with control (and just call it control, because technically it is not a treatment) – this is just a small suggestion, not really important

L 107-110: I would maybe move the 2.3. Statistical Analysis subsection at the end of Materials and Methods (as last subsection)

L 176-177: “chlorine had a low concentration in the leaves, and there were no differences in the salt concentration” – Can you maybe explain this because it is really surprising? I was wondering if the chloride determination in the aqueous extract by titration with silver nitrate is an adequate method for chloride detection in plant tissue; from my experience, the concentration was usually to low for titration method as it is not precise or sensitive enough. Maybe the method used camouflaged the results, or maybe you can provide other explanation…

L 223: Table 4 – I think something went wrong with the width of the first and the sixth column

Good luck! :-)

Author Response

(The authors gave the same response as above.)

Round 2

Reviewer 1 Report

I cannot change my recommendation from major revisions at this time. The authors did augment the conclusions section with final conclusions, and it did help the paper. However, the main suggestion of adding discussion and comparing to past work was not addressed. The manuscript still has potential to add to the body of literature, however as the suggestions were not addressed, I cannot change my recommendation.

Furthermore, no response to review was included supporting the lack of change or justifying the lack of changes, so I have no information as to why the suggestions weren't followed.  I recommend for future manuscripts, the authors include a point-by-point response to review, justifying the changes made or why the changes were not made. This would greatly help the manuscript and facilitate discussion between the authors and reviewers. 

Author Response

Dear Reviewers,

Thank you for your time and your suggestions to improve of the paper.

Best regards,

Gulom Bekmirzaev
